# Dynamics and Structure Formation of Confined Polymer Thin Films Supported on Solid Substrates

**DOI:** 10.3390/polym13101621

**Published:** 2021-05-17

**Authors:** Mujib Ur Rahman, Yonghao Xi, Haipeng Li, Fei Chen, Dongjie Liu, Jinjia Wei

**Affiliations:** 1State Key Laboratory of Multiphase Flow in Power Engineering, Xi’an Jiaotong University, Xi’an 710049, China; mujeeb715@gmail.com; 2School of Chemical Engineering and Technology, Xi’an Jiaotong University, Xi’an 710049, China; xiyoga2018xjtu@163.com (Y.X.); Li_haipeng@stu.xjtu.edu.cn (H.L.); feichen@mail.xjtu.edu.cn (F.C.)

**Keywords:** polymer thin film, solvent rinsing, viscosity, dewetting

## Abstract

The stability/instability behavior of polystyrene (PS) films with tunable thickness ranging from higher as-cast to lower residual made on Si substrates with and without native oxide layer was studied in this paper. For further extraction of residual PS thin film (h_resi_) and to investigate the polymer–substrate interaction, Guiselin’s method was used by decomposing the polymer thin films in different solvents. The solvents for removing loosely adsorbed chains and extracting the strongly adsorbed irreversible chains were selected based on their relative desorption energy difference with polymer. The PS thin films rinsed in chloroform with higher polarity than that of toluene showed a higher decrease in the residual film thickness but exhibited earlier growth of holes and dewetting in the film. The un-annealed samples with a higher oxide film thickness showed a higher decrease in the PS residual film thickness. The effective viscosity of PS thin films spin-coated on H-Si substrates increased because of more resistance to flow dynamics due to the stronger polymer–substrate interaction as compared to that of Si-SiO_x_ substrates. By decreasing the film thickness, the overall effective mobility of the film increased and led to the decrease in the effective viscosity, with matching results of the film morphology from atomic force microscopy (AFM). The polymer film maintained low viscosity until a certain period of time, whereupon further annealing occurred, and the formation of holes in the film grew, which ultimately dewetted the film. The residual film decrement, growth of holes in the film, and dewetting of the polymer-confined thin film showed dependence on the effective viscosity, the strength of solvent used, and various involved interactions on the surface of substrates.

## 1. Introduction

As new technologies are introduced to enable the manufacturing of ever-smaller devices, it has been shown that in the past two decades, the use of polymer nanometer films and structures underwent rapid growth, such as thin-film transistors, semi-conductor chips, biosensors, and organic photovoltaics [1]. The control of morphology and stability of polymer thin film on solid substrates is of paramount importance, especially for new emerging polymer-based nanotechnologies. For example, block copolymers, offering an attractive patterning technology because of the self-assembling into various morphologies, are considered to undergo extensive studies for wide applications, including biomolecules adsorption, nanolithography, nanoparticle synthesis, and high-density information storage media [2,3,4,5,6].

However, polymer thin films, especially under confinement (when the film thickness reaches a value smaller than the unperturbed radius of gyration (R_g_)), often exhibit distinguishable micro and macro characteristics (microstructures, stability, rheological properties, etc.) from the bulk [7,8,9,10]. Spin coating is a well-established and widely used rapid technique to deposit uniform polymer thin film on a flat substrate in a controlled manner. The residual stress induced by polymer spin coating is often considered the main reason for polymer thin film instabilities and micro and macro properties change since the non-equilibrium stressed chain conformations form after the rapid evaporation of solvent [11,12]. To remove the residual stress and obtain the equilibrated polymer chain conformations, the thermal annealing process at a temperature over the glass transition temperature (T_g_) for a longer time than the bulk reptation time is typically required [13].

However, the films will become unstable with the formation of holes and voids either with or without annealing work and will show dewetting behavior when the film thickness reaches a very small value (i.e., closed to R_g_) [14,15]. Chen et al. [8] reported that unstable PS thin films form cylindrical holes on the solid substrate after a certain time of annealing, and the growth speed of the radius of the hole was related to polymer viscosity. Therefore, in this scenario, it becomes of interest to study how one can predict and improve the thermal stability of the thin films by controlling the dewetting phenomena. Previous literature has found that the polymer thin film stability on a solid surface may depend on factors including polymer–substrate interaction, polymer film thickness, viscosity, molecular weight, and temperature [11,16,17,18,19,20,21]. A study by Xue et al. [22] discussed a method to improve the stability of polymer films by adjusting the composition of the film with the addition of some additives, new end-groups, or cross-linking agents.

However, these approaches are more difficult to set up for ultra-thin films. New promising strategies based on the view of spin-coated film structure need to be focused and expanded to inhibit the dewetting of such ultra-thin films without any modification in the composition [23,24,25,26,27,28,29]. When adsorbed onto a solid substrate, the macromolecules will form a film with two different regions depending upon the history of processing [28]. By using the measurements of second-harmonic generation (SHG), Rotella et al. [30] verified that at the early stage of chain adsorption, the mean orientation of monomers changed and lied parallel to the substrate with more polymer–substrate contacts, and the layer showed a high density, which is defined as the strongly adsorbed chains at the inner region. Then, the later arriving chains were defined as loosely adsorbed chains and free chains (no contacts) at the outer region since the monomers showed less substrate contacts and remained perpendicular. Solvent rinsing, as described by Guiselin [26], is an effective method to rinse the polymer thin films by solvent, which can remove the loosely adsorbed chains and extract the volume fraction profile of irreversible adsorbed polymer thin film for further characterization and study. Recently, Bal et al. [31] used the top-down approach to remove the outer loosely adsorbed polymer chains by rinsing in toluene and obtained a stable ultra-thin polymer film on an oxide-free substrate, as expected from van der Waals interaction theory, and a dewetted film on an oxide substrate. They attributed the stability of film on the oxide-free substrate to the capability of toluene as a good solvent to remove residual stress from the film and the density variation inside the film. Jiang et al. [32] used the strong solvent chloroform to remove more loosely adsorbed chains and obtained a strongly adsorbed inner layer. Moreover, they used a different annealing time and different substrates to study the polymer–substrate interactions on the polymer-confined films. Koga et al. [24] reported that after 120 days of putting the samples in toluene solvent, they obtained a non-zero high-density inner layer with thickness independent of polymer molecular weight, while the thickness of outer loosely adsorbed layer showed dependence on molecular weight, and residual thickness increased with higher polymer molecular weight. As for rinsing in solvents with a higher polarity, it was observed that the film instability increased and showed earlier dewetting. Chen et al. [33] found that a longer equilibration time was required for THF-prepared samples because the adsorbed layers of these samples evolved more slowly and showed a bigger change in viscosity with annealing due to out of equilibrium than those of the toluene-prepared samples. Therefore, solvents with much higher polarity should be avoided, and nonpolar solvents, such as toluene and chloroform, are preferred in such top-to-down rinsing work. Jiang et al. [34] reported that the time growth of the flattened layers composed of the three different polymers (polystyrene, poly (2-vinylpyridine), and poly (methyl methacrylate)) exhibited similar power-law growth thermal annealing by X-ray reflectivity experiments, which have similar inherent stiffness and bulk glass transition temperature but different affinities with Si substrates, also facilitating polymer adsorption from the melt on planar silicon (Si) substrates. Seemann et al. [35] studied the effect of long-range and short-range forces on the stability of PS on different types of substrates. According to the theory of interfacial energy, the film stability of the medium air/PS/SiO_x_/Si system strongly depended upon the oxide layer thickness. Thus, H-Si and SiO_x_-Si are two typical substrates that can be used to study the impact on the polymer–solid interaction.

Based on the polymer/solid interface by the irreversible adsorption, these polymer chains in confined thin film experience a significant change in local viscosity relative to their bulk counterpart [36,37,38,39,40]. Chen et al. [41] studied the effect of viscosity and surface-promoted slippage of PS thin films on solid substrates. They reported that unstable polymer thin films formed cylindrical holes on the solid substrate after a certain time of annealing, and the growth speed of the radius of holes was related to polymer viscosity. The outer layer at the polymer/air interface has higher mobility and enhanced chain dynamics, whose properties differ from that of the inner layer at the polymer/solid interface since in the inner layer the dynamics are reduced due to the presence of a strongly adsorbed polymer layer [41,42,43,44]. By controlling this adsorption kinetics, the structural and dynamic properties, such as glass transition temperature, viscosity, segmental mobility, crystallization, wettability, and the thermal expansion coefficient of polymer thin films on solid substrates, could be tailored easily. These are versatile properties of confined polymer thin films, but stability remains the main issue to deal with in such conditions.

In this work, a series of PS films with thickness ranging from higher as-cast to lower residual on H-Si and Si-SiO_x_ substrates were prepared to reveal how polymer/substrate interaction affects the desorption. Residual film thickness, the amount of loosely adsorbed polymer chains, and the morphology of the thin films were observed and supposed to be tunable according to the results of rinsing and annealing. The formation of holes in PS film after decreasing the film thickness by rinsing and annealing at a different roughening time was observed, and the cross-sectional profiles of holes were presented. The effective viscosity for two different substrates was also calculated and discussed in order to facilitate the understanding of the chain dynamics and properties and control of the adsorption kinetics in polymer/substrate interfaces based on the AFM method.

## 2. Materials and Methods

### 2.1. Materials

PS with different M_w_ (12–2300, kg/mole) and poly-dispersity indices (M_w_/M_n_ = 1.01–1.1) was purchased from Sigma-Aldrich (Shanghai, China). The raw Si wafers (001) were purchased from Henan Yixin Co., Ltd. (Zhengzhou, China). The solvents and cleaning agents, toluene (C_6_H_5_CH_3_) > 99.5%, chloroform (CHCl_3_) > 99.8%, sulfuric acid (H_2_SO_4_) 98%, hydrogen peroxide (H_2_O_2_) 35 wt%, and hydrogen fluoride (HF) were purchased from Sigma-Aldrich (Shanghai, China).

### 2.2. Sample Preparation

#### 2.2.1. Chemical Treatment for Substrate Cleaning

The raw Si (001) wafers were cut into 1.5 cm × 1.5 cm slides, pre-cleaned by deionized water, and then submerged in a hot piranha solution (i.e., a mixture of H_2_SO_4_ and H_2_O_2_ by the ratio of 7:3) to remove the organic contamination (caution: the piranha solution is extremely dangerous and highly corrosive when in contact with the skin or eyes and is an explosion hazard when mixed with organic chemicals/materials; extreme care should be taken when handling the solution). The slides should be put in the mixture and kept under fume hood at 130 °C for 10 min and then rinsed with deionized water thoroughly, followed by drying with 99.99% nitrogen. The slides were further cleaned in oxygen plasma for 20 min, whereupon they were ready for spin coating. The Si substrate after piranha treatment was represented as Si-SiO_x_. The SiO_x_ surface was predominantly covered by OH groups and showed a hydrophilic behavior. The oxide layer thickness was measured by spectroscopic ellipsometry SE-VE (Wuhan Eoptics Technology Co., Ltd., Wuhan, China) and found to be 2 nm.

Hydrogen fluoride (HF) etching was performed to remove the oxide layer; a Teflon beaker was used since HF corroded the glassware. Clean oxide Si substrates after piranha treatment were immersed in an aqueous solution of 2% HF for 20 s to remove the native oxide layer and then dried by 99.99% nitrogen. The submersion in HF solution resulted in the replacement of OH by Si−H surface groups. An oxide layer of about 1.3 nm in thickness was found even after the HF treatment, which was possibly due to the atmospheric oxygen and moisture, as reported previously [25,32,45,46]. With HF treatment, the oxide layer was removed, and the H-terminated hydrophobic Si substrate (H-Si substrate) was obtained.

#### 2.2.2. Spin Coating

The spin coating technique was performed for PS thin film deposition. Solutions with different mass concentrations in anhydrous toluene (99.8%) were prepared, and an aging time of 24 h was given for the better dissolution of PS. The solutions were filtered through a PTFE membrane with a pore size of 0.1 *μ*m before spin coating. The spin coater (Zhengzhou Instrument Co., Ltd., Zhengzhou, China) was used, and the PS solution was dispensed on the substrate and rotated at a selected speed for 35 s. The PS final film thickness depends upon the nature of the PS solution, i.e., viscosity and solvent evaporation rate and the parameters of spin rotation and time.

#### 2.2.3. Solvent Rinsing

The PS samples were prepared by spin coating of different M_w_ solutions on H-Si, and Si-SiO_x_ substrates were kept for a whole night before annealing and rinsing. Then, some of these samples were pre-annealed while some were not. To extract the interfacial polymer structures near the substrate surface, we followed Guiselin’s approach [21] to unveil the irreversibly adsorbed inner polymer layer with all the samples rinsed in two solvents for different periods of time (from 10 s to 90 min) and dried by a mild flow of N_2_ at ambient temperature and pressure. Then, the PS residual film thickness was monitored by ellipsometry. The schematic illustration of spin coating and rinsing of PS chains is shown in Figure 1. As summarized by Rotella et al. [30], at the early stage of adsorption, early arriving chains on the solid substrate have more surface-segment contacts as compared to late arriving chains, especially after pre-annealing work (increase the contacts number at the inner region), which are defined as the strongly adsorbed chains (green). The later arriving chains are defined as loosely adsorbed chains (red) with less substrate contacts, and the free chains (blue) have no contacts with substrate.

The quality of solubility for the solute in the solvent is determined by relative energy difference. The solvent with a smaller relative energy difference has a higher solubility ratio. When a good solvent is used in rinsing of homogeneous polymer thin film, it not only diffuses into polymer film but also plasticizes the polymer chains at the polymer–substrate interface. This plasticization favors the chain disentanglement in the presence of solvent and ultimately removes them with the solution [23,47,48,49]. The outer layer of loosely adsorbed polymer chains decomposes as the sample is inserted in a good solvent like toluene. This decomposition of the loosely adsorbed layer occurs due to the effect of entropic force induced by exposing this layer to an environment free from polymer chains [50].

Toluene has a lower value of polarity and higher solubility with PS, and it can easily remove polymer chains by promoting disentanglement [51,52,53]. After removing the non-adsorbed and loosely adsorbed chains, the solvent is then directly in contact with strong adsorbed chains, which have greater surface contacts and cannot be removed unless using some strong solvents. Thus, in this work, fresh chloroform was used for further removal of adsorbed chains with a strong solvent, residue films [46], until no obvious change in the thickness was observed. It was found that for a prolonged rinsing of the samples after 90 min of rinsing, the quality of the PS film declined with an increase in roughness at the top surface, while the film thickness did not appreciably change. This selective extraction of the adsorbed layer method was feasible owing to the large difference in the desorption energy between the PS inner and outer loosely adsorbed chains [54,55].

### 2.3. Characterization

Spectroscopic ellipsometry is commonly used in laboratories for the thickness measurement of the adsorbed layer. This technique depends on the changes in the polarization state of the light after reflecting from a thin film. The thicknesses of these PS thin films at five different positions were checked by ellipsometry with incidence angles of 65°, as showed in Figure 2a. The films’ thicknesses were determined by fitting the Ψ and Δ spectra using the Cauchy model of the refractive index (n) as a function of wavelength (*λ*) for *λ* from 400 to 850 nm with Equation (1) [56]:(1)$$n\left(\lambda \right)=A+\frac{B}{\lambda^{2}}+\frac{C}{\lambda^{4}}$$

A multilayer model with substrate Si, SiO_2,_ and PS was used. To reduce the number of free parameters, the thickness of the oxide layer was determined before deposition of the organic layer, and then, this thickness of the oxide layer was fixed as the host material thickness. For example, the result of one PS451k film spin coated on SiO_x_-Si substrate using this technique is shown in Figure 2b. The thicknesses in five locations are 30.34 nm, 29.61 nm, 29.75 nm, 30.09 nm, and 29.89 nm, where the average thickness is 29.94 nm and the standard deviation is 0.29 nm, which is acceptable.

The film thickness was also confirmed by atomic force microscopy (AFM) considering the height of film steps obtained by removing the organic layer with a soft scratch. The tip used had a resonant frequency of 300 kHz and a spring constant of 40 N/m. The scan rate was 0.5–1 line per second with a scanning density of 512 lines per frame. Figure 2c shows the AFM morphology of the same film checked before by ellipsometry. In the location marked with “scratch” in Figure 2c, part of the film is scratched by a soft razor blade, and the topographic figure is shown in Figure 2d, leaving some PS material remaining at the edge of the scratch with a bright color. Then, the area marked by a red frame was selected for further analysis. In Figure 2e, it is shown clearly by the scratch that it exposes the substrate (black), and a small PS material remained at the edge of the cut (white) and PS film (brown). The measurement results (29.54 nm) of the distance between the substrate and film show an agreement with those by ellipsometry (29.94 nm). Several groups of film thickness were assessed using both of the approaches were to make mutual confirmation, and ellipsometry was determined to be the preferred method due to the advantages of optical non-contact and quick measurement.

AFM is a powerful method for providing more information than film thickness in different modes. Chen et al. [41] studied the effect of viscosity and surface promoted slippage of PS thin films on solid substrates based on AFM results in tapping mode. Nieswandt et al. [6] reviewed a variety of mechanisms for gaining control over block copolymer order as well as many of the applications of these materials, and peak force quantitative nano-mechanical mapping (QNM) experiments on the Bruker AFM proved to provide interesting information regarding the different mechanical responses of the substrates. In a similar way, AFM can also work for different film thicknesses along with the provision of a correlation with rheological results and information on mechanical stability. In this work, AFM (Bruker Corp., Billerica, MA, USA) in the tapping mode operation was mainly used to study the surface morphology of the films. The morphology of the PS film with the growth of the hole is shown in Figure 3.

The surface topography of the films was also measured by tapping-mode AFM at times *t* in the initial stage where linear analyses were valid. All of the topographic data were Fourier transformed and processed to give the power spectral density (PSD), *A_q_^2^*(*t*), as detailed by Chen et al. [41]. Adopting an adiabatic approximation, we analyzed the ensemble-averaged quasi-equilibrium elastic vibrations. For the vibrations, we assume [3*μ_0_*/(2*h*_0_^3^*q*^2^)]|*u***_q_**|^2^ for the elastic energy associated with individual modes with wave vector *q* and amplitude *u***_q_**; *q* ≡ |*q*| and *μ_0_* is the shear modulus of the film. A linear stability calculation assuming the lubrication approximation (*qh*_0_ << 1) and stable films gives:(2)$$A_{q}^{2}\left(t\right)=A_{q}^{2}\left(0\right)\exp\left(2{\Gamma \prime }_{q}t\right)+\left(\frac{k_{B}T}{\gamma q^{2}+G^{\prime \prime }\left(h_{0}\right)}\right)\left(1-\exp\left(2\Gamma \prime_{q}t\right)\right)$$
where $\Gamma \prime_{q}=-M_{\mathrm{tot}}q^{2}{\left[{\left(\gamma q^{2}+G^{\prime \prime }\left(h_{0}\right)\right)}^{-1}+{\left(\frac{3\mu_{0}}{h_{0}^{3}q^{2}}\right)}^{-1}\right]}^{-1}$.

In Equation (2), Γ’*_q_* is the relaxation rate of the modes, *k*_B_ is the Boltzmann constant, *T* is absolute temperature, is surface tension, and *G(h_0_)* is the van der Waals potential of the film. Equation (2) predicts that there is a characteristic time, τ, for viscoelasticity of the film. Below this characteristic time, the film surface evolves like that of elastic solid with a shear modulus of *μ*_0_. However, for t ≫ τ, the surface of the film evolves like viscous liquid with viscosity *η_eff_* ≡ *h_0_^3^*/(3*M_tot_*) and characteristic time τ = *η_eff_*/*μ*_0_.

## 3. Results and Discussion

Residual film thicknesses and AFM results of PS451k films on different substrates in toluene, with or without annealing work, are given in Figure 4.

The annealed samples show a smaller film thickness decrease in the residual film as compared to un-annealed samples in Figure 4a. As-cast polymer thin films have residual stress due to rapid evaporation of the solvent during the film preparation. Annealing of the film above T_g_ can reduce the stress and cause the polymer chains to become relaxed. Annealing promotes adsorption and increases the number of polymer–surface contacts and the film thickness of the adsorbed layer, which leads to a smaller decrease in film thickness in the residual film. The AFM results for the morphology of annealed and un-annealed samples are shown in Figure 4b–e. The rapid solvent evaporation during spin coating leaves some free volume on the substrate interface and by annealing the film, more activation energy is provided to PS chains to cover this free volume [34]. By removing these empty spaces with adsorption of the PS chains, the density of the film increases at the substrate interface.

In addition, we can observe from Figure 4a that different substrates also influence the residual film thickness. This is because the chains’ adsorption and segmental dynamics largely depend on the kind of substrate used. The substrate roughness of Si-SiO_x_ is higher than that of H-Si substrate, leading to stronger segmental mobility restriction at the Si-SiO_x_ interface as compared to the H-Si interface [57]. PS film becomes saturated as the solvent molecules are diffused in the film, and it reduces the glass transition temperature, which causes segmental mobility in the film. Thermal fluctuations are amplified even at room temperature, causing spontaneous dewetting in the film [58,59].

Regarding the type of solvents used for leaching work, the morphology of PS451k thin films spin coated on SiO_x_ substrates is depicted in Figure 5. It is clear that the film thickness reduces with the increase in the solvent rinsing time, the film undergoes dewetting, especially when h_resi_ is close to the transition (a regime from unstable to metastable) PS film thickness of about 2.9 nm predicted by the effective interfacial potential Φ(h) [31,60]. In the insets, fast Fourier transform (FFT) images indicate the instability in the film. The stability and morphology of the residual polymer thin films strongly depend on the confined polymer film thickness. When polymer film is leached with a good solvent, toluene, for example, its morphology turns from a homogeneous to a bi-continuous structure at a film thickness of 2.5 nm after 30 min of rinsing. However, when the polymer film is rinsed with strong solvent chloroform, the film morphology changes from a homogenous to a bi-continuous structure just after 10 min of rinsing. This can be explained by solvent-driven dewetting where strong polar solvent plays a significant role in the dewetting of the polymer thin film [61].

PS thin films, which are stable during thermal annealing on oxide-free substrates, exhibit dewetting with rinsing, even in the poor solvent of acetone [58]. Lee et al. [58] proposed that thermal dewetting was caused by instability in the film due to the long-range force of van der Waals interactions, and solutal dewetting was caused by the short-range force of polar interactions.

Theoretically, the effective interfacial potential Φ (h) has been widely used for the prediction of confined polymer thin film stability on oxide and oxide-free substrates. Seemann et al. [42] studied the effect of long-range and short-range forces on the stability of PS on different types of substrates. The effective interfacial potential (Φ) for PS thin films on the silicon substrate (Si/SiO_x_/PS/Air) is given in Equation (3).
(3)$$\Phi \left(h\right)=\frac{{-A}_{{\mathrm{SiO}}_{x}}}{{12\pi h}^{2}}+\frac{A_{{\mathrm{SiO}}_{x}}{\;-\;A}_{\mathrm{Si}}}{12\pi {\left({h+d}_{{\mathrm{SiO}}_{x}}\right)}^{2}}+\frac{C}{h^{8}}$$
where h is the film thickness; A_Si_ and A_SiOx_ are Hamaker constants for Si/PS/Air and SiO_x_/PS/Air systems; d_SiOx_ is oxide thickness; and C is the short-range interaction parameter. In the study of the nonlinear theory of film rupture, William and Davis [60] noted that when the second derivative of effective interfacial potential with respect to a value was negative, the system was unstable. When its value was higher than zero, it was metastable, and the transition film thickness from unstable to metastable occurred at 2.9 nm film thickness. However, this theory did not validate a conflicting result found between a stable 5 nm PS residual film and a 5 nm unstable as-cast PS film [31,61]. According to the theory of interfacial energy, the film stability of the medium air/PS/SiO_x_/Si system strongly depended upon oxide layer thickness. To study the effect of SiO_x_, we therefore used H-Si and SiO_x_-Si substrates. For a given thickness of PS film, the value of effective interfacial potential Φ(h) followed the order HF etched Si > native oxide Si >> thermal oxide Si [58].

To investigate the thermal stability of PS thin films, samples were annealed at 150 °C for different periods of time in a vacuum (~10^−3^ m bar). At very short annealing times, some well-dried samples revealed a pattern typical of spinodal decomposition as seen in AFM topographic results, and the surface roughness was also excessively large, so these samples were not considered in the analysis of film kinetics. The dynamics for the growth of the hole in a film is only observable if the density of the hole is not high to the extent that they must have a space to grow in the uniform film.

In Figure 6a–c, the AFM results are shown for the PS451k film prepared on Si-SiO_x_ substrate. The topographic image of the as-cast PS film is shown in Figure 6a with a uniform film. The PS sample was annealed in a vacuum oven for 5 min at a temperature of 150 °C. The AFM image in Figure 6b shows the uniform film without any appearance of the hole for 5 min of thermal annealing. The surface roughness of the film before annealing was ~3 Å; however, it increased to 4 ± 1 Å, which may be due to the thermal excitation of capillary wave fluctuations after annealing [27]. When the sample was annealed for 15 min at the same temperature, the holes grew in the film, as shown in Figure 6c, and the depth of holes is shown in Figure 3b,c. The radius of the hole increased, leading to hole coalescence.

In Figure 7a–e, the AFM figures are for PS451k film samples on H-Si substrates when annealed at a temperature of 150 °C for a longer time. Figure 7a shows the as-cast PS451k sample with uniform and smooth film. In Figure 7b, the AFM image shows the increased number and size of holes for the same PS451k sample. With the increase in annealing time, the size and number of the holes kept on increasing. The film’s structure became bi-continuous when annealed at the same temperature for a longer annealing time of 4 h in the vacuum oven, as shown in Figure 7c. This bi-continuous structure then changed to isolated droplets upon further annealing, as shown in Figure 8d,e, after annealing for 16 h and 24 h, respectively, at 150 °C.

For samples of PS451k thin film on Si-SiO_x_, the morphological changes for multistep thermal annealing are shown in Figure 8a–e. Figure 8a shows the smooth film with no holes, and the remaining figures show the dewetted film. After 20 min of annealing, the film showed a large number of dewetting holes with sizes larger than those of H-Si substrates, and the holes became large as the annealing time increased, as shown in Figure 8b. According to the literature [62], this result suggests that a considerable amount of molecular scale porosity exists in the film. The molecular scale porosity present in the film would then decrease the glass transition temperature and enhance the segmental mobility of PS chains, which can further lead to the dewetting of the film. The film turned into a bi-continuous structure, a typical spinodal-like dewetting morphology, after 4 h of annealing, as shown in Figure 8d. After 24 h of annealing, the bi-continuous structures broke into isolated droplets, as shown in Figure 8e.

The effective interfacial theory does not consider the stress present in the film, and this could be the main reason for instability in the film. Thomas et al. [63] reported that the stresses in the chain conformation could be removed for a shorter timescale, but recovery of bulk viscosity took more than 100 h. Several experiments found that as-cast films with high M_w_ are out of equilibrium and exhibit residual stress and reduced chain entanglement [11,23,24,32]. Complete re-entanglement of chains is required for the recovery of bulk viscosity, which takes a longer time [63]. Reiter in his experiment predicted that by decreasing film thickness, the viscosity of the PS thin film should decrease from bulk viscosity and cause the dewetting of the film [64]. Chen et al. [8,41] studied the confinement effect on the viscosity and surface-promoted slippage of PS thin films on solid substrates. To measure the viscosity of polymer thin films on solid substrates, dewetting properties could be used [19,65]. Unstable polymer thin films form cylindrical holes on the solid substrate after a certain time of annealing, and the growth speed of the radius of those holes is related to polymer viscosity.

Here, we calculated the PSD and viscosity of the film as described in Equation (3). Figure 9a is the viscosity (η) of 3 nm PS film on Si-SiO_x_ substrates (black) and H-Si substrates (red) at 120 °C. In Figure 9b, η of the 8 nm PS film on two different substrates at 120 °C is shown. As seen, the two sets of data overlap for the PS molecular weight of M_w_ < 60 kg/mole. Above this molecular weight, the viscosity of PS films on H-Si substrates surpasses the viscosity of PS films on Si-SiO_x_ substrates. Figure 9c shows the viscosity of pure PS450k film at 172 °C. Thicker films had viscosity similar to that of the bulk value.

The chain segments at the free surface of the PS thin film have a higher mobility than the other layers in the system [66]. By decreasing film thickness, the overall effective mobility of the film increased, leading to a decrease in the effective viscosity and T_g_. Consequently, the film ruptured via the formation of holes. The velocity of hole growth increased largely with a decreasing film thickness that was consistent with decreasing viscosity by decreasing film thickness. In annealing experiment of PS451k residual film on Si-SiO_x_ substrate, the observed growth of holes, bi-continuous structure and isolated droplets in the film at different annealing time proved to be well matched with the results discussed above.

## 4. Conclusions

In this paper, the effect of annealing and solvent rinsing on PS residual film was studied. Two different solvents, toluene and chloroform, were used for the extraction of PS residual thin film on two different substrates. H-Si substrate has a stronger interaction with PS as compared to Si-SiO_x_ substrates due to higher interfacial potential Φ(h), showing higher wetting stability and residual film. The samples rinsed in high-polarity solvent exhibited faster dissolution and desorption of PS chains. This desorption behavior of PS residual chains showed dependence on the polymer–solvent relationship as well as on polymer–substrate interaction. Regarding the morphology of samples, the chloroform-rinsed samples exhibited earlier dewetting in the films due to their higher polarity than toluene. By analyzing the 2D FFT, a preferred wavelength for dewetting was observed for samples with a thickness less than the transition film thickness, which validates the results with effective interfacial potential theory.

Annealing promotes PS chain adsorption, and the decrease in PS residual film thickness was less for these samples. The samples annealed for a shorter time showed uniform morphology, which may be due to the relaxation of PS chains after the hole nucleation observed in the film. The holes coalesced and turned into bi-continuous and ultimately isolated droplets as the annealing time increased.

The effective viscosity for two different substrates was calculated and was found to be similar for small M_w_, presuming that enhanced flow dynamics at the free surface dominated the flow dynamics in the whole film. However, as M_w_ increased, the flow dynamics changed, and the substrate with stronger attraction showed more resistance to flow dynamics. Meanwhile, the effective viscosity also decreased as the polymer film thickness decreased and deviated from the bulk viscosity for sufficiently small film thickness.

## Figures and Tables

**Figure 1 polymers-13-01621-f001:**
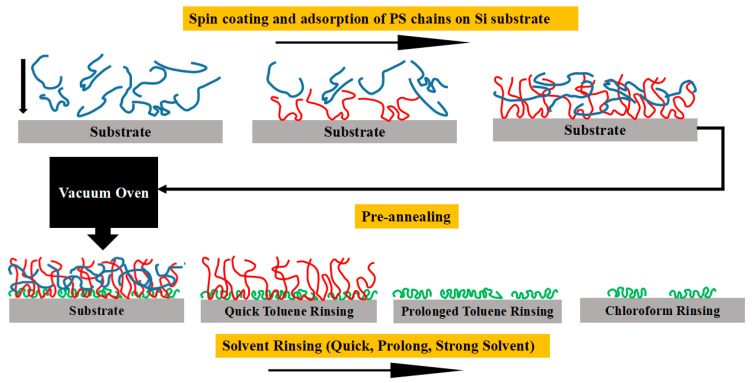
Schematic illustration of PS spin coating and formation of the adsorbed layer, vacuum annealing, and solvent leaching and acquirement of residual film. Color scheme: blue, free chains; red, loosely adsorbed chains; green, strongly absorbed chains.

**Figure 2 polymers-13-01621-f002:**
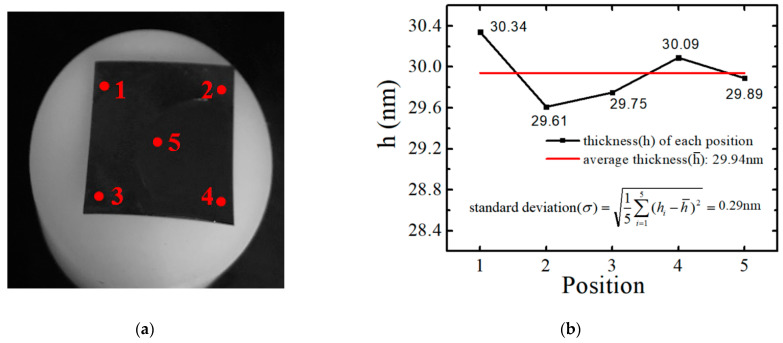
Results of assessing the thickness of one PS451k film spin coated on SiO_x_-Si substrate using two techniques. (**a**) Schematic illustration of assessing films thicknesses at five different positions by ellipsometry; (**b**) results of ellipsometry: the thickness of five positions, average thickness, and standard deviation; (**c**) AFM topographic figure of the film; (**d**) AFM topographic figure of the film scratched on the area marked by a white line in (**c**); (**e**) results of assessing film thickness by AFM, with a comparison of the heights of the substrate and film of the inset illustration, which comes from the area marked by a red frame in (**d**).

**Figure 3 polymers-13-01621-f003:**
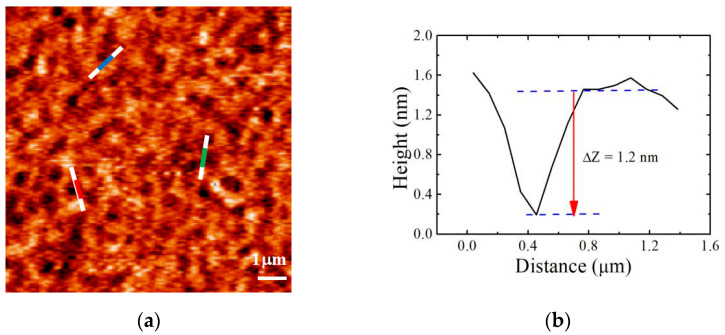
(**a**) AFM image of the appearance of holes in 11 nm PS451k film on SiO_x_ substrate annealed at 150 °C for 15 min; (**b**–**d**) cross-sectional profiles of (**a**), labels of color: blue (**b**), red (**c**), green (**d**).

**Figure 4 polymers-13-01621-f004:**
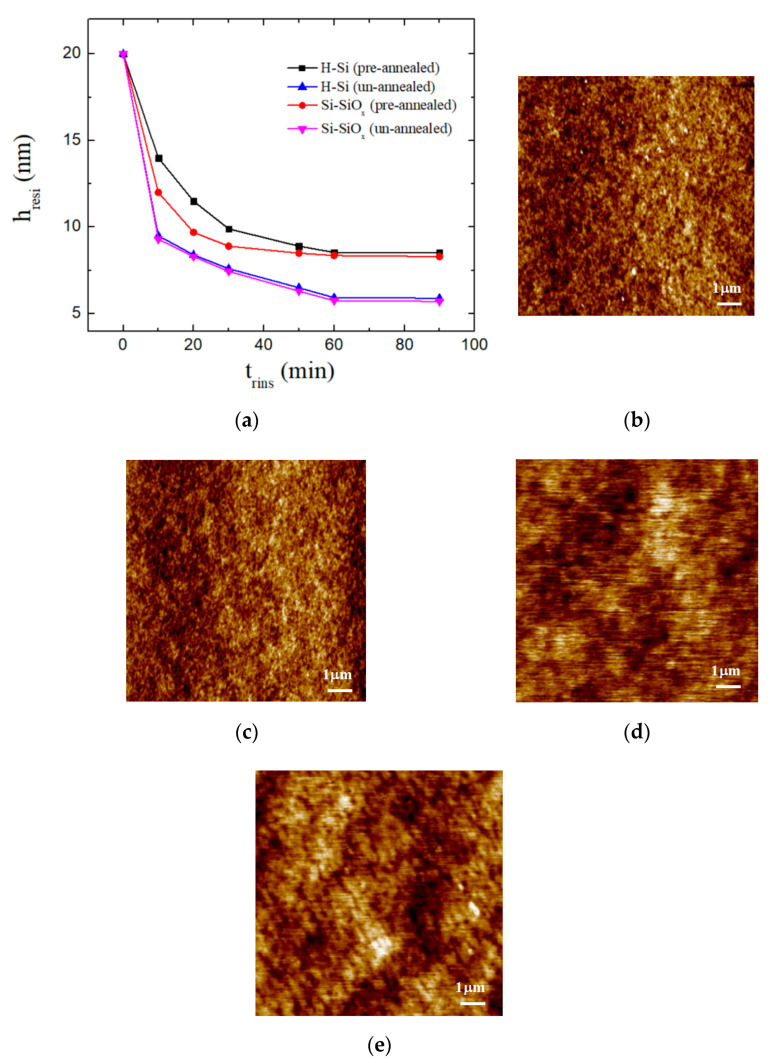
Residual film thickness (**a**) and AFM results (**b**–**e**) of PS451k films on H-Si and Si-SiO_x_ substrates in toluene, with or without annealing work. (**b**) H-Si (pre-annealed); (**c**) H-Si (un-annealed); (**d**) Si-SiO_x_ (pre-annealed); (**e**) Si-SiO_x_ (un-annealed).

**Figure 5 polymers-13-01621-f005:**
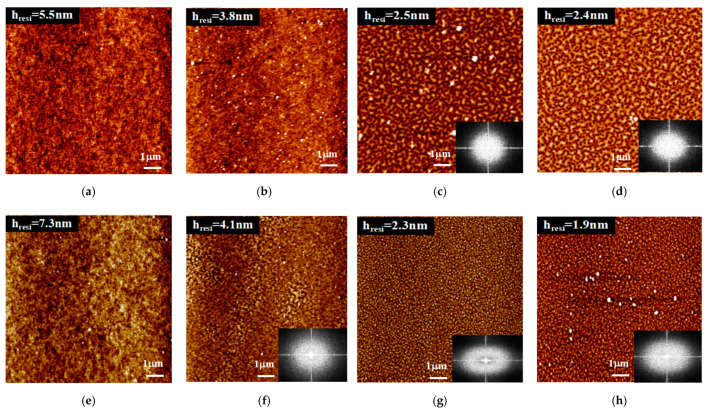
AFM images for PS451k spin coated on Si-SiO_x_ substrates rinsed in toluene and chloroform solvents: (**a**–**d**) rinsed in toluene for 1 min, 10 min, 30 min, and 60 min; (**e**–**h**) rinsed in chloroform for 1 min, 10 min, 30 min, and 60 min. In the insets, 2D FFT images show the instability in the films.

**Figure 6 polymers-13-01621-f006:**
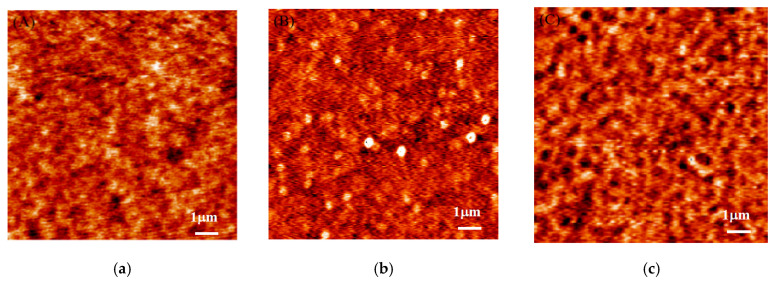
AFM surface morphological evolution of 11 nm PS451k thin film prepared on the Si-SiO_x_ substrate during the thermal annealing process. The scan sizes of the AFM height images are 10 µm × 10 µm. (**a**) AFM image of PS451k with smooth and continuous film on Si-SiO_x_; (**b**) the same sample annealed at 150 °C in a vacuum oven for 5 min; (**c**) the same sample when annealed at 150 °C for 15 min.

**Figure 7 polymers-13-01621-f007:**
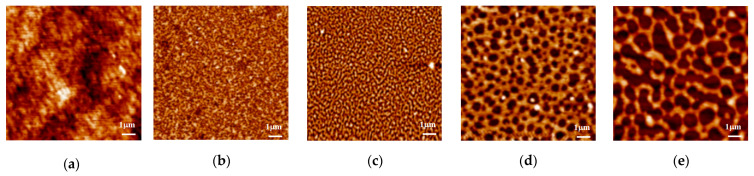
AFM surface morphological evolution of 20 nm PS451k thin film prepared on the HF-etched Si substrates during the multistep thermal annealing process for a longer time. The scan sizes of the AFM height images are 3 µm × 3 µm. (**a**) AFM image of PS451k with a smooth and continuous film on H-Si; (**b**) the sample annealed at 150 °C in a vacuum oven for 1 h; (**c**) the sample when annealed at 150 °C for 4 h; (**d**) the sample annealed at 150 °C for 16 h; (**e**) the sample when annealed at 150 °C for 24 h.

**Figure 8 polymers-13-01621-f008:**
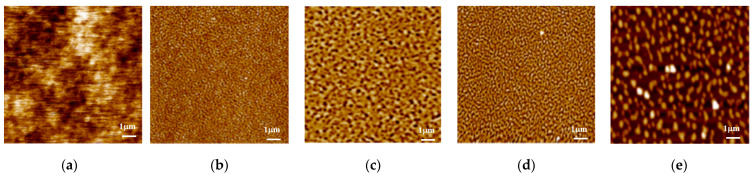
AFM surface morphological evolution of 20 nm PS451k thin film prepared on the Si-SiO_x_ substrates during the multistep thermal annealing process for a longer time. The scan sizes of the AFM height images are 3 µm × 3 µm. (**a**) AFM image of PS451k with a smooth and continuous film on Si-SiO_x_; (**b**) the sample annealed at 150 °C in a vacuum oven for 1 h; (**c**) the sample when annealed at 150 °C for 4 h; (**d**) the sample annealed at 150 °C for 16 h; (**e**) the sample when annealed at 150 °C for 24 h.

**Figure 9 polymers-13-01621-f009:**
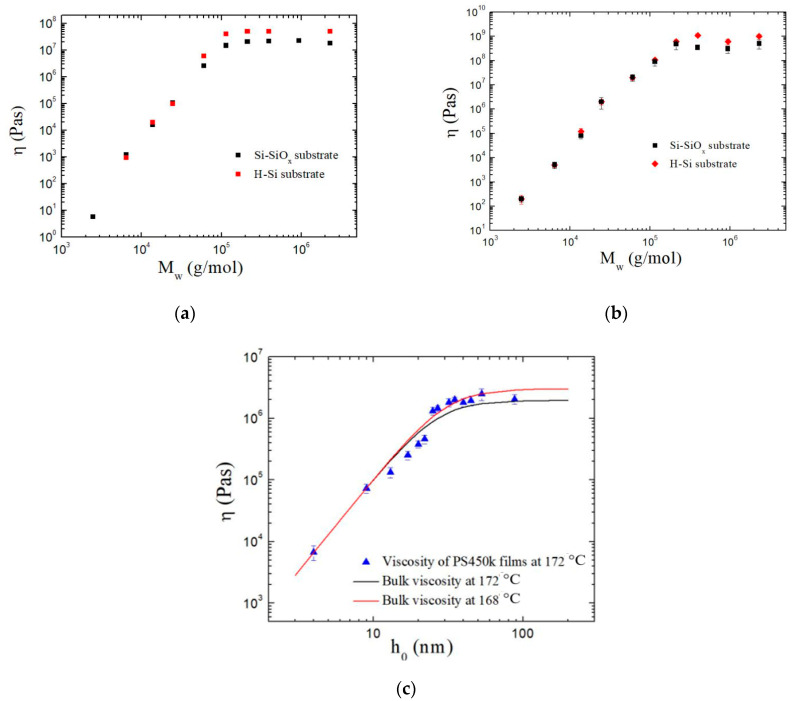
The effective viscosity (η) for different M_w_ of PS on different substrates with film thickness of (**a**) 3 nm and (**b**) 8 nm. (**c**) The η of pure PS451k in different thicknesses.

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
