# Peer review of "Dynamics and Structure Formation of Confined Polymer Thin Films Supported on Solid Substrates"

_polymers, 2021, doi:10.3390/polym13101621_

Round 1
Reviewer 1 Report
The manuscript entitled: "Dynamics and structure formation of confined polymer thin films supported on solid substrates" is interesting and has some potential for publication in the journal polymer. However, some revisions are necessary.
- Figure 1. is a little complex to understand and the major issue that I have is since you use the same Mw for the PS to coat the substrate, why the different colors and sizes? or did the authors spin coat blends of different PS?
- Why only Chloroform and Toluene? These two solvents have a polarity difference but none of them is a polar solvent. a polar solvent used could be THF.
- Figure 2 has some type of mistake. Please correct properly. Specifically, what is 2b showing and why it is in that aspect ration given? Please include this in the figure caption.
- Ellipsometry is a strong characterization method for thin-film formation. I would like to see some statistics for the thickness measurements and how the uniform is the surface.
- Why the authors tried only PS?
- Did the authors consider using also different substrates, e.g., mica?\
- Polymer thin-films is a major part of nanotechnology for decades. there are a lot of works describing such methodologies. Especially block copolymers have been considered as very unique examples and also extensive studies with AFM have been done. This is a part missing from the introduction and the authors can add. Here are a few examples that can be used https://doi.org/10.1039/D1PY00074H, https://doi.org/10.1021/acs.macromol.7b02218, https://doi.org/10.1002/admi.201901580, https://doi.org/10.1016/j.mser.2004.12.003, etc.
- Was any possible for quantitative nanomechanical mapping experiments with the AFM? (Peak Force QNM method on the Bruker AFMs)? This method can provide fantastically information regarding the different mechanical responses of the substrate. See for example here https://doi.org/10.3390/ma12193145, for block copolymer. In a similar way can work also for different thicknesses. this provides also a correlation with rheological results and provided information on mechanical stability.
- All the AFM images are missing the scale bar (x, y, z scale) information. please provide the necessary information.
- Finally and most critical, the authors should express the reason and the aim for publishing this work. On what topic is this work helpful? Express clearly the novelty!
Reviewer 2 Report
The manuscript entitled "Dynamics and structure formation of confined polymer thin films supported on solid substrates" deals with the characterization of the behavior of PS thin films on different substrates.
In my view, the work deserves to be published on Polymers, since the topic is very interesting, the experimental procedure is well designed and the conclsions are supported by obtained results. Furthermore, the manuscript is well arranged and easy to follow.
Therefore, I recommend the publication of the manuscript as it stands.
